# Integrated millimeter-wave cavity electro-optic transduction

Kevin K. S. Multani [1,2,3,5] ✉, Jason F. Herrmann [1,4,5] ✉, Emilio A. Nanni [3] & Amir H. Safavi-Naeini [1,4] ✉

Emerging communications and computing technologies will rely ever-more on expanding the useful radio frequency spectrum into the millimeter-wave and terahertz frequency range. Both classical and quantum applications would benefit from advancing integration and incorporation of millimeter-wave and electro-optic technologies into common devices, such as modulators. Here we demonstrate an integrated triply-resonant, superconducting electro-optic transducer. Our design incorporates an on-chip 107 GHz niobium titanium nitride superconducting resonator, modulating a thin-film lithium niobate optical racetrack resonator operating at telecom wavelengths. We observe a maximum photon transduction efficiency of $\eta_{OE} \approx 0.82 \times 10^{-6}$ and an average single-photon electro-optic interaction rate of $g_0/2\pi \approx 0.7$ kHz. We also present a study and analysis of the challenges associated with the design of integrated millimeter-wave resonators and propose possible solutions to these challenges. Our work paves the way for further advancements in resonant electro-optic technologies operating at millimeter-wave frequencies.

Next-generation communications, imaging, and sensing technologies rely on harnessing millimeter-wave (mm-wave) and terahertz (THz) frequencies. Occupying a spectral region between electronics and optics, the millimeter-wave (30–300 GHz, 10–1 mm) and THz (0.1–10 THz, 3000–30 μm) frequencies offer advantages such as increased data transmission rates[1,2], improved imaging resolution[3,4], new radar and ranging modalities[5], and access to new regimes of physical phenomena[6]. Millimeter-wave frequencies offer several compelling advantages for quantum science applications beyond conventional microwave systems. Superconducting processors that operate at higher frequencies are able to operate at elevated temperatures, allowing access to greater cooling power[7]. This could potentially reduce the scaling barriers for fault-tolerant quantum computing[8]. Building on these developments, there is a nascent effort to create quantum devices in the millimeter-wave range, as demonstrated by recent breakthroughs in the demonstration of mm-wave superconducting qubits[9,10]. This emerging quantum toolbox includes innovations such as neutral atom systems serving as quantum-enabled millimeter-wave-to-optical

transducers with high conversion efficiency[11], mm-wave electromechanical[12] and optomechanical systems[13,14], and mm-wave phononic bandgap structures that couple to defects in diamond[15]. An important missing element is an integrated transduction mechanism between millimeter-wave and photonic systems, essential for networking these devices and enabling hybrid quantum systems.

Frequency conversion and transmission of information between disparate systems are essential in classical and quantum information and sensing systems. Although classical signals can be addressed with commercially available modulators, quantum signals require specialized transducers. Quantum transducers, with their significantly different figures of merit and requirements, have garnered significant attention in the past decade. In that time, integrated quantum transducers interconverting between microwave (3 GHz) and optical (193.5 THz) frequencies have been developed using direct electro-optic[16–21], piezo-, electro-, and optomechanical approaches[22–27].

In this work, we present a triply-resonant mm-wave cavity electro-optical platform based on thin-film lithium niobate (TFLN). We provide

[1]E.L. Ginzton Laboratory, Stanford University, Stanford, CA, USA. [2]Department of Physics, Stanford University, Stanford, CA, USA. [3]SLAC National Accelerator Laboratory, Stanford University, Menlo Park, CA, USA. [4]Department of Applied Physics, Stanford University, Stanford, CA, USA. [5]These authors contributed equally: Kevin K. S. Multani, Jason F. Herrmann. ✉e-mail: kmultani@stanford.edu; jfherrm@stanford.edu; safavi@stanford.edu

details regarding the superconducting mm-wave cavity and TFLN optical cavity, presenting crucial analysis needed to achieve the quantum operation of such mm-wave devices. Leveraging integrated photonics and RF co-packaging, our device is capable of mm-wave-to-telecom direct electro-optic transduction. A fully fledged device could link systems operating at elevated frequencies and temperatures[7,8]. Examples of such systems include superconducting qubits[10], neutral atoms[11], and long-baseline telescopes[28,29]. Additionally, such a transducer may also be used in multistage transduction to link current quantum hardware at microwave frequencies to optical frequencies, via a mm-wave intermediary[30–32].

## Results

### Device overview and operating principle

Our device utilizes the electro-optic effect, where a mm-wave signal with frequency $\Omega$ modulates the refractive index of a region of X-cut TFLN, through the electro-optic coefficient, $r_{33} \approx 31\ \mathrm{pm\,V}^{-1}$[33]. In the presence of an optical pump ($\omega_p$), the modulation produces red- and blue-detuned sidebands ($\omega_p \pm \Omega$) via three-wave mixing. By matching the frequency of the mm-wave resonant mode ($\omega_{RF} \approx \Omega$) with the free spectral range (FSR) of the optical cavity (i.e., the difference between a telecom optical pump mode $\omega_0$ and the next detuned mode of the optical cavity, $\omega_\pm = \omega_0 \pm \omega_{RF}$), we form a triply-resonant cavity electro-optic system (see Fig. 1a), which enables efficient pump photon utilization. When the input and generated fields are resonant with their respective cavities, the on-chip quantum efficiency for photon transduction in the low-cooperativity (cooperativity

$C \equiv 4g_0^2 n_{c,0}/(\kappa_+ \kappa_{RF}) \ll 1$) regime can be written as,

$$\eta_{OE} = \left| \frac{\alpha_+^{out}}{\beta^{in}} \right|^2 \approx 4C \left( \frac{\kappa_{e,+}}{\kappa_+} \right) \left( \frac{\kappa_{e,RF}}{2\kappa_{RF}} \right), \tag{1}$$

where $|\alpha_+^{out}|^2 = P_+^{out}/(\hbar\omega_+)$ is the (blue) optical photon flux generated on-chip, and $|\beta^{in}|^2 = P_{RF}^{in}/(\hbar\omega_{RF})$ is the mm-wave photon flux incident to the chip. In these expressions, $n_{c,0}$ is the intracavity photon population of the optical pump mode, $\kappa_{ej}$ ($\kappa_j$) is the external (total) coupling rate of mode $j$, and $P_j^{in/out}$ is the on-chip input/output power of mode $j$, where $j \in \{+, -, 0, RF\}$. The quantity $g_0$ is the single-photon electro-optic coupling rate, which fundamentally sets the strength of the interaction. Note, due to our coupling approach for the mm-wave cavity, the coupling efficiency ($\kappa_{e,RF}/2\kappa_{RF}$) for the mm-wave resonator can at most be 50%.

To implement the triply-resonant system described above, we integrate a superconducting mm-wave cavity made from a thin film of niobium titanium nitride (NbTiN) with a photonic racetrack resonator made from TFLN atop a sapphire substrate (Sa). We choose NbTiN because it has a much higher superconducting transition temperature $T_c$ and shorter quasiparticle lifetime than aluminum, making it suitable for mm-wave operation. As mentioned above, the mm-wave modulation produces both red- and blue-detuned sidebands, which are resonant with the optical racetrack modes. Although the presence of the red sideband is not detrimental in this work, it would be undesirable during single-photon quantum operation (see Supplementary

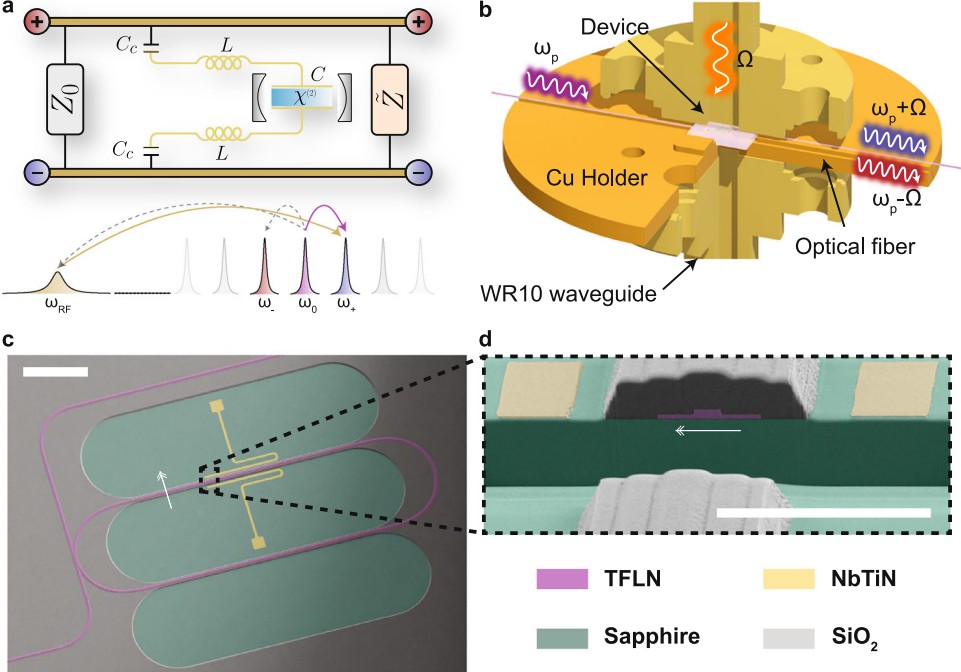

**Fig. 1 | Device operating principle and overview. a** Illustration of the lumped element model of the transducer. The incoming mm-wave fields are carried by a transmission line with characteristic impedance $Z_0$ and are coupled to the superconducting resonant circuit (colored in gold) by capacitance $C_c$. The outgoing mm-wave waves experience a different characteristic impedance $\widetilde{Z}$, due to the Sa substrate. The intracavity electric field of the mm-wave mode drops across a capacitor of capacitance $C$, which surrounds the LN, depicted as a medium with $\chi^{(2)}$ non-linearity. The mirrors enveloping the LN crystal indicate that the optical modes are resonant. The cartoon below the circuit model depicts the triply-resonant three-wave mixing (beam splitter) interaction realized by our device, where $\omega_{RF}$, $\omega_0$, and $\omega_+$ are the mm-wave mode, the optical pump mode, and the up-converted blue-detuned optical mode frequencies. For completeness, the transparent gray arrows

show the entangled pair generation process. **b** Rendering of the transducer device and packaging using Shapr3D. The highlighted arrows show the directionality of the optical pump (purple, $\omega_p$), mm-wave pump (orange, $\Omega$), and up-converted optical sideband (blue, $\omega_p + \Omega$), respectively. We also show the red sideband, $\omega_p - \Omega$ photons, because in our device, both processes occur. The integrated photonic and mm-wave circuitry lies atop a translucent sapphire substrate and is mounted in the center of the copper holder. **c** False-colored scanning electron micrograph of the full device. Note, we etch the oxide cladding so that the NbTiN electrodes sit directly atop the substrate (reducing mm-wave loss). Scale bar indicates 100 μm. **d** Zoomed in cross-section of a section of the interaction region. Scale bar indicates 10 μm. Double arrowhead indicates the LN crystal-$z$ axis.

Information S1.1). To address this issue in future devices, one could engineer the local dispersion of the racetrack resonator around the optical modes of interest, in order to suppress one of the modulation sidebands. Possible approaches include coupling a second optical resonator to the first at 3× the FSR of the original racetrack, or making use of alternative mode mixing schemes[16,17,34].

We refer to the region where the TFLN racetrack resonator is surrounded by superconducting electrodes as the "interaction region." The superconducting electrodes form a coplanar stripline resonator where the fundamental mode frequency, $\omega_{RF}$, is set by the total electrode length (see "Methods": "Superconducting millimeter-wave cavity design"). A key aspect of our mm-wave cavity design is that it maintains a unipolar electric field along the interaction region (to avoid cancellations in modulation), without crossing over the integrated photonic components to avoid optical losses due to TFLN proximity to metal and mm-wave RF losses due to electrode proximity to oxide[35]. A false-colored SEM of the device and an etched cross-section of the interaction region are presented in Fig. 1c, d.

We design a custom copper package to simultaneously address the device with both mm-wave and optical fields. The packaging, shown in Fig. 1b, interfaces with both WR10 rectangular waveguides and SMF-28 optical fibers. The optical fibers are aligned and glued to the on-chip grating couplers, maintaining their alignment from 298 to 4 K[36,37]. We mount WR10 rectangular waveguides to the copper package so that the in-waveguide mm-wave field is normally incident on the device. The on-chip superconducting electrodes pick up the incident field, like an antenna, and subsequently modulate the electric field in the interaction region.

## Optical and millimeter-wave cavity spectroscopy

We mount the fully packaged device inside a cryostat and perform separate spectroscopic measurements for the optical and mm-wave cavity resonances at a base temperature of roughly 4.9 K. Beginning with the optics, we record a transmission spectrum of the optical racetrack resonator by inputting and scanning a tunable laser tone through the device, and recording the transmitted optical signal on a photodetector, depicted in Fig. 2a. By calibrating the wavelength axis of this spectrum using a fiber Mach-Zehnder interferometer (MZI) with known FSR, we infer an optical resonance spacing of $|\omega_\pm - \omega_0| \approx 2\pi \times 105.25$ GHz. Furthermore, we observe a ~3 dB decrease in transmission through the fibers and gratings after the base temperature is reached, which we attribute to temperature-induced changes in the device refractive indices, leading to shifted peak frequency responses of the grating couplers.

Using an off-chip electro-optic modulator (EOM) and vector network analyzer (VNA), we perform heterodyne measurements to further characterize each optical mode[38,39]. These measurements let us directly infer the external and total coupling rates, $\kappa_{e,j}$ and $\kappa_j$, of each mode. In this measurement, we lock the pump laser blue-detuned from each mode and then, by driving the EOM with the VNA source port, sweep a sideband across the optical resonance. The chip's output is collected on a high-speed photodiode, and the beat tone between the locked pump and the swept sideband is recorded on the receiving VNA port. We fit an input-output model to the phase response of this measurement, from which we can infer the amplitude response of the signal (for additional details regarding this measurement and fitting model, please see "Methods": "Experimental setup," Supplementary Fig. 6, and Supplementary Information S4.2 and S4.2.1). Our observed amplitude and phase response for the optical pump resonance are shown in Fig. 2b, c, respectively. We summarize the inferred optical mode parameters from these measurements in Table 1.

We characterize the mm-wave cavity using a VNA with frequency extension modules to measure the response from 75 to 115 GHz. To identify superconducting features, we perform a temperature sweep by changing the platform temperature of the cryostat and measuring

the mm-wave spectrum at each temperature. We observe a broad feature that red-shifts and increases in linewidth with increasing temperature, consistent with the expected response of a superconducting cavity[40]. We identify this feature as the mm-wave cavity response (see Fig. 2d, e). The top two panels of Fig. 2d depict the measured mm-wave transmission spectrum. Importantly, the expected Lorentzian lineshape of a bare mm-wave cavity resonance is difficult to distinguish due to the presence of many avoided crossings (apparent hopping in Fig. 2d). These crossings indicate that the superconducting mm-wave cavity resonance is hybridizing with extraneous, narrow linewidth modes, the frequencies of which are not temperature dependent. We identify these extraneous modes to be bulk electromagnetic resonances supported within the Sa substrate. As such, we refer to these modes as "substrate modes." In short, at mm-wave wavelengths, the Sa substrate acts as a dielectric cavity that supports many modes that can hybridize with the bare superconducting cavity mode (see "Methods": "Substrate mode modeling and analysis" and Supplementary Fig. 2).

It is important to model the substrate modes' hybridization with the superconducting mm-wave mode because they affect our inference of key figures of merit. Therefore, we developed a multi-parameter input-output model of the millimeter-wave transmission spectrum (see Eqs. 6 and 7 in "Methods": "Substrate mode modeling and analysis" and see Supplementary Fig. 5). Using this model, we fit the mm-wave transmission spectra we measure and extract from the results the parameters of the bare superconducting mode, substrate modes, and relevant coupling rates (see Supplementary Information S1.3, S1.4, S2, and S3 for details on our model and fitting procedure). A set of three mm-wave spectra at different temperatures, along with fits to our multiparameter model, are shown in Fig. 2e (see Table 1 for fitted mode parameters at 4.9 K).

## Triply-resonant millimeter-wave electro-optic transducer characterization

We characterize the device transduction efficiency $\eta_{OE}$ and the single-photon coupling rate $g_0$ by separately varying both the optical pump power and mm-wave modulation power. Each optical pump power yields a different intracavity photon number in the optical mode $\omega_0$. Millimeter-wave modulation at frequency $\omega_{RF} \approx |\omega_\pm - \omega_0|$ then generates optical sidebands that are measured directly using an optical spectrum analyzer (OSA). An example OSA trace is shown in Fig. 3a. For each optical pump power, we also vary the mm-wave modulation power, thereby obtaining multiple measurements using the OSA. From each OSA measurement, we determine the on-chip photon flux in both the blue sideband and the superconducting mm-wave mode. We thereby obtain a linear plot of the on-chip blue sideband photon flux, $\dot{n}_+$, versus the incident mm-wave photon flux, $\dot{n}_{RF}$. The slope of this line, $\eta_{OE} = \dot{n}_+ / \dot{n}_{RF}$, is the electrical-to-optical number conversion efficiency, and constitutes a single data point in Fig. 3b. More details of this procedure are available in Supplementary Information S2 and S3. Crucially, this modulation also produces a red-detuned sideband at $\omega_-$, as seen in the OSA data (Fig. 3a).

Note that for each optical pump power setting, it is important that we lock our laser close to the optical pump-mode resonance frequency, $\omega_0$ (see "Methods": "Pump detuning calibration and locking"), so that the optical cavity is sufficiently populated to drive the three-wave mixing interaction. We maintain this lock via PID feedback control, where we adjust the optical pump laser's output wavelength with a voltage applied to a piezoelectric actuator. The optical transmission through the chip is then measured on a photodiode and used as the feedback signal for our PID loop. This PID control is implemented with a Red Pitaya (see Supplementary Fig. 6, Supplementary Information S4.4). Fluctuations in the optical transmission and between optical power settings can vary the detuning between the locked laser and the optical resonance frequency, so we fit this detuning as a parameter in post-processing (see "Methods": "Pump detuning calibration and

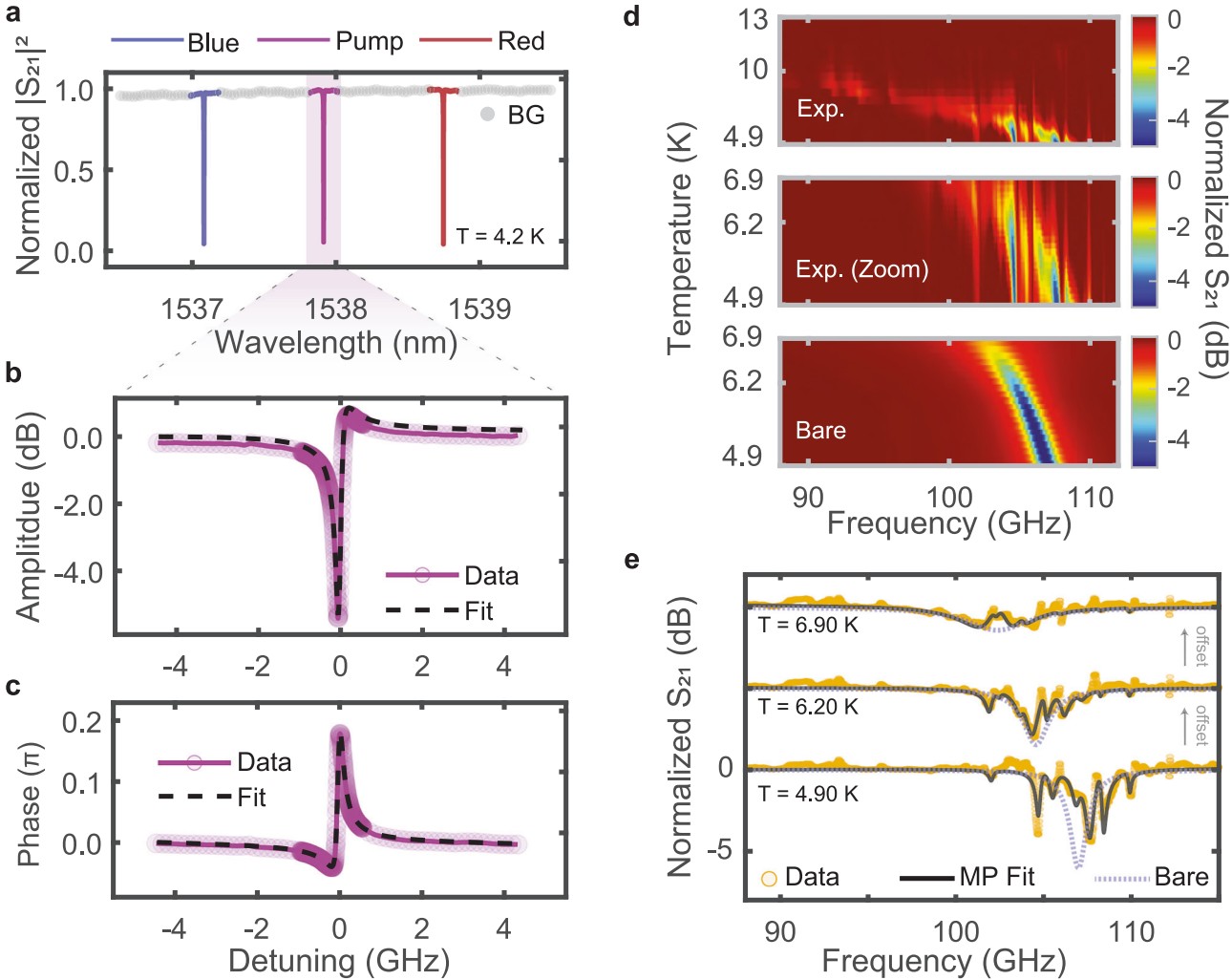

**Fig. 2 | Cryogenic characterization of optical and mm-wave cavities.**
**a** Normalized transmission of the three optical modes relevant in our experiments. Purple indicates the pump mode, $\omega_0$, blue (red) indicates the blue- (red-) detuned sideband mode, $\omega_+$ ($\omega_-$). While the distance from $\omega_0$ to the red or blue modes is slightly different due to dispersion, both are within a few MHz of 105.25 GHz. **b** Self-heterodyne amplitude response of the optical pump mode, measured on a VNA. We only show the pump mode here for brevity, but we perform these measurements for the two sideband modes as well. **c** Self-heterodyne phase response of the optical pump mode, measured on a VNA. We fit the phase response to infer the optical mode resonant frequencies $\omega_j$, external coupling rates $\kappa_{e,j}$, and the total loss rates $\kappa_j$. These parameters are then used to plot the fits depicted in (**b**, **c**). **d** Millimeter-wave transmission spectrum as a function of cryostat platform temperature. The data are normalized to data where $T > T_c \approx 10$ K. The top panel depicts a broad superconducting feature, red-shifting and broadening as temperature increases. The

response is riddled with many avoided crossings coming from the superconducting mode hybridizing with high-$Q$ modes of the Sa substrate (substrate modes). The middle panel shows a region of the data where 4.9 K < $T$ < 7 K. The bottom panel shows the bare response of the mm-wave cavity, inferred from a multiparameter fit (see "Methods": "Substrate mode modeling and analysis," Supplementary Information S1.3 and S1.4). **e** Normalized transmission at three temperatures depicting the measured data (gold circles), multiparameter fit (solid black line), and bare mm-wave response (solid light blue line). As the temperature increases (the offset is 5 dB), the mm-wave cavity red-shifts and broadens. The broadening indicates an increase in internal loss rate, observed as an increase in the minimum of the Lorentzian lineshape (reduction in contrast). For $T \approx 4.9$ K the inferred bare mm-wave cavity parameters are $(\omega_{RF}, \kappa_{RF}, \kappa_{e,RF})/(2\pi) = (106.99, 1.674, 0.419)$ GHz (see Table 1 for all device parameters).

locking" and Supplementary Information S2.2). Furthermore, we lock the laser wavelength slightly blue-detuned from the optical resonance to increase stability against thermo-optic and photorefractive mode drift (see Supplementary Information S5 for more details on these effects).

In the low-cooperativity limit, the on-chip transduction efficiency should increase linearly with the intracavity optical photon number $n_{c,0}$, as indicated by Eq. (1). We observe the expected linear trend at low intracavity pump photon numbers. However, as the intracavity photon number (i.e., optical pump power) increases, we see a deviation in the efficiency curve (Fig. 3b). We attribute this effect to quasiparticle generation in the superconducting cavity electrodes, resulting from the absorption of optical pump photons. These quasiparticles affect the superconducting cavity in a manner analogous to a temperature

increase, an effect discussed in further detail below (see Fig. 4). As a result, we can use our multiparameter model to capture the nonlinear deviation in the efficiency curve, by including the optical-power-dependent red-shift and linewidth broadening of the mm-wave cavity (see Optically induced quasiparticle generation and Supplementary Information Sections S3 and S3.2).

Inferring the single-photon coupling rate $g_0$ from the measured transduction efficiency hinges on an accurate estimate of the on-chip photon numbers in both the optical and mm-wave cavities (see Eq. 9 in "Methods": "Inferring the electro-optic coupling rate $g_0$"). Achieving this requires careful calibration of the losses in the optical and mm-wave RF measurement paths, along with precise power calibration at cryogenic temperatures. Even minor misalignments, temperature drifts, or repeated unplug/replug cycles can lead to shifts in the optical

**Table 1 | Device parameters as inferred from independent measurements, described in the "Results" section**

| Cavity mode | Frequency $\omega_k/2\pi$ (THz) | Total loss $\kappa_k/2\pi$ (GHz) | Extrinsic loss $\kappa_{e,k}/2\pi$ (MHz) |
|---|---|---|---|
| $\omega_O$ | 194.932 | 0.201 | 88.1 |
| $\omega_+$ | 195.037 | 0.210 | 87.3 |
| $\omega_-$ | 194.827 | 0.199 | 85.8 |
| $\omega_{RF}$ | 0.106993 | 1.674 | 838 |

| Substrate mode | Frequency $\omega_n/2\pi$ (GHz) | Total loss $\kappa_n/2\pi$ (MHz) | Coupling $J_n/2\pi$ (MHz) |
|---|---|---|---|
| $\omega_1$ | 102.163 | 302.06 | 779 |
| $\omega_2$ | 102.579 | 260.11 | 235 |
| $\omega_3$ | 103.746 | 189.03 | 0.63 |
| $\omega_4$ | 104.978 | 160.01 | 682 |
| $\omega_5$ | 105.843 | 365.68 | 857 |
| $\omega_6$ | 107.022 | 445.42 | 455 |
| $\omega_7$ | 108.180 | 122.75 | 440 |
| $\omega_8$ | 109.784 | 182.93 | 467 |

| Transduction parameter | Symbol | Value | Unit |
|---|---|---|---|
| EO coupling rate (inferred from efficiency) | $g_0^\eta/(2\pi)$ | 707 | Hz |
| EO coupling rate (inferred from sideband ratio) | $g_0^\Upsilon/(2\pi)$ | 685 | Hz |
| Maximum measured efficiency (on-chip) | $\eta_{OE}^{max}$ | $0.82 \times 10^{-6}$ | — |
| Single-photon cooperativity | $C_O$ | $6 \times 10^{-12}$ | — |
| Maximum measured cooperativity | $C$ | $1 \times 10^{-5}$ | — |

The optical measurements are taken when no RF power is applied. The mm-wave substrate mode parameters are identified from a fit procedure discussed in the Supplementary Information (Section 3). The $J_n$ values reported here correspond to coupling between the substrate mode and the mm-wave cavity mode at base temperature (4.9 K).

and mm-wave power delivered to the chip, which in turn directly affect the inferred values of $g_0$. We therefore perform cryogenic calibrations of both the optical and mm-wave paths to minimize these sources of error, as described in the Supplementary Information Sections S1.1, S2.1 and S4.

To validate the estimate of the electro-optical coupling rate from the conversion efficiency discussed above, we also estimate $g_0$ using the converted sideband ratio (SBR). The SBR is defined as $\Upsilon_{+,0} \equiv P_+/P_0$, where $P_0$ ($P_+$) refers to the optical pump power (blue sideband power), measured by the OSA. Because the SBR relies on taking the ratio of two optical powers that traverse the same optical path, the SBR value, and therefore our estimate of $g_0$ from the SBR, is less sensitive to deviations of the optical path transmission from our calibrated loss values. On resonance, the SBR can be written as

$$\Upsilon_{+,0} \approx \frac{4g_0^2}{\kappa_+^2} \cdot \frac{4\kappa_{e,0}\kappa_{e,+}}{(\kappa_0 - 2\kappa_{e,0})^2} \cdot n_{c,RF}, \qquad (2)$$

as with $\eta_{OE}$, but instead of scaling with the number of intracavity optical pump photons $n_{c,0}$, the SBR scales with the number of intracavity mm-wave photons $n_{c,RF}$ (see Eq. 10 in "Methods": "Inferring the electro-optic coupling rate $g_0$" and Supplementary Information S1.4 for more details). The measured powers in the blue and red sidebands, highlighted by the horizontal dashed lines in Fig. 3a, are not equal. The blue sideband power is −72.25 dBm, while the red sideband power is −74.35 dBm, corresponding to a blue-red sideband ratio of $\Upsilon_{+,-} = P_+/P_- \approx 1.62$. We attribute this power imbalance to frequency detunings between the optical pump wavelength, the mm-wave modulation

frequency, and the frequencies of the two distinct optical resonances, $\omega_\pm$. This power discrepancy is additionally affected by differences in $\kappa_e$ and $\kappa$ between the optical modes, as described in Supplementary Information Section S1.1. From our multiparameter input-output model that includes the substrate modes, we estimate $\Upsilon_{+,-} \approx 1.68$, indicating good agreement between our theory and the measurement (see Supplementary Information Section S1.1).

The inferred values of $g_0$ from both methods for low pump power (corresponding to the linear region in Fig. 3b) are shown in Fig. 3e. In principle, the inferred $g_0$ should be constant for each pump power, for both methods. Experimental factors such as polarization drift, inaccuracies in path calibration for the optics and RF, and uncertainties in modal rates ($\kappa$'s) and detunings can cause discrepancies. In our case, we find fairly good agreement across all pump powers, indicating that our measurements are consistent within the statistical uncertainty. If we average all the estimates of $g_0$ from the $\eta_{OE}$ data, we obtain $g_0^\eta = 2\pi \cdot 707(681, 733)$ Hz, where the parentheses indicate the 95% confidence interval. Likewise, for the estimates using the SBR, we obtain $g_0^\Upsilon = 2\pi \cdot 685(645, 722)$ Hz.

Lastly, we characterize the bandwidth of the transducer by sweeping the modulation frequency ($\Omega$) for each optical and mm-wave RF power. Representative data from bandwidth measurements are shown in Fig. 3d. We observe a maximum conversion efficiency at $\Omega = 2\pi \times 105.285$ GHz ($\Delta = 0$), in agreement with the measurement of the MZI-calibrated FSR discussed above, $|\omega_+ - \omega_0| = 2\pi \times 105.25$ GHz. Together with the data in Fig. 3d, we plot two sets of dashed lines. The long-dashed line depicts a Lorentzian model, with a peak value chosen to match the data and a linewidth chosen to correspond to the measured optical mode linewidth of the blue sideband mode at $\omega_+$. Similarly, the short-dashed line depicts a Lorentzian model of the bare mm-wave resonance, again with the peak chosen to match the data and with a linewidth corresponding to the inferred superconducting mm-wave resonance linewidth. Note that there is a small side peak in the conversion efficiency at $\Delta \approx -2\pi \times 0.6$ GHz, which we attribute to hybridization with a substrate mode (see Eq. 9 in "Methods": "Inferring the electro-optic coupling rate $g_0$").

In contrast to previous work on optomechanical and electro-optic transducers, our conversion bandwidth is limited by the optical cavity linewidth, instead of the mm-wave (or mechanical) cavity linewidth. This puts us in the so-called reversed dissipation limit of cavity electro-optics[41]. In this regime, electro-optically induced amplification or cooling of an optical mode could be realized if the cooperativity is on the order of unity, $C = 4(g_0^2 n_{c,0})/(\kappa_0 \kappa_{RF}) \approx 1$. However, in this work, we measure a maximum of $C \approx 10^{-5}$, and therefore, the amplification or cooling effects are inaccessible with our current device.

**Optically induced quasiparticle generation**

We attribute the nonlinear behavior in efficiency to optically induced quasiparticle generation in the superconducting mm-wave cavity (see Fig. 3b). As the number of optical pump photons increases, an increasing number of optical photons are scattered into the local cryostat environment and eventually are absorbed by the superconducting mm-wave electrodes. This absorption breaks Cooper pairs in the superconducting electrodes, increasing the non-equilibrium quasiparticle density. An increase in the density of quasiparticles degrades the performance of the superconducting cavity in a manner analogous to increasing the device operating temperature: dissipation increases, thereby increasing the cavity loss rate; and inductance increases, thereby lowering the resonance frequency[40].

The quasiparticle number density can be expressed as,

$$n_{qp}(T_{eff}) = \left(2N(0)\sqrt{2\pi k_B T_{eff}\Delta(T_{eff})}\right)\exp\left(\frac{\Delta(T_{eff})}{k_B T_{eff}}\right), \qquad (3)$$

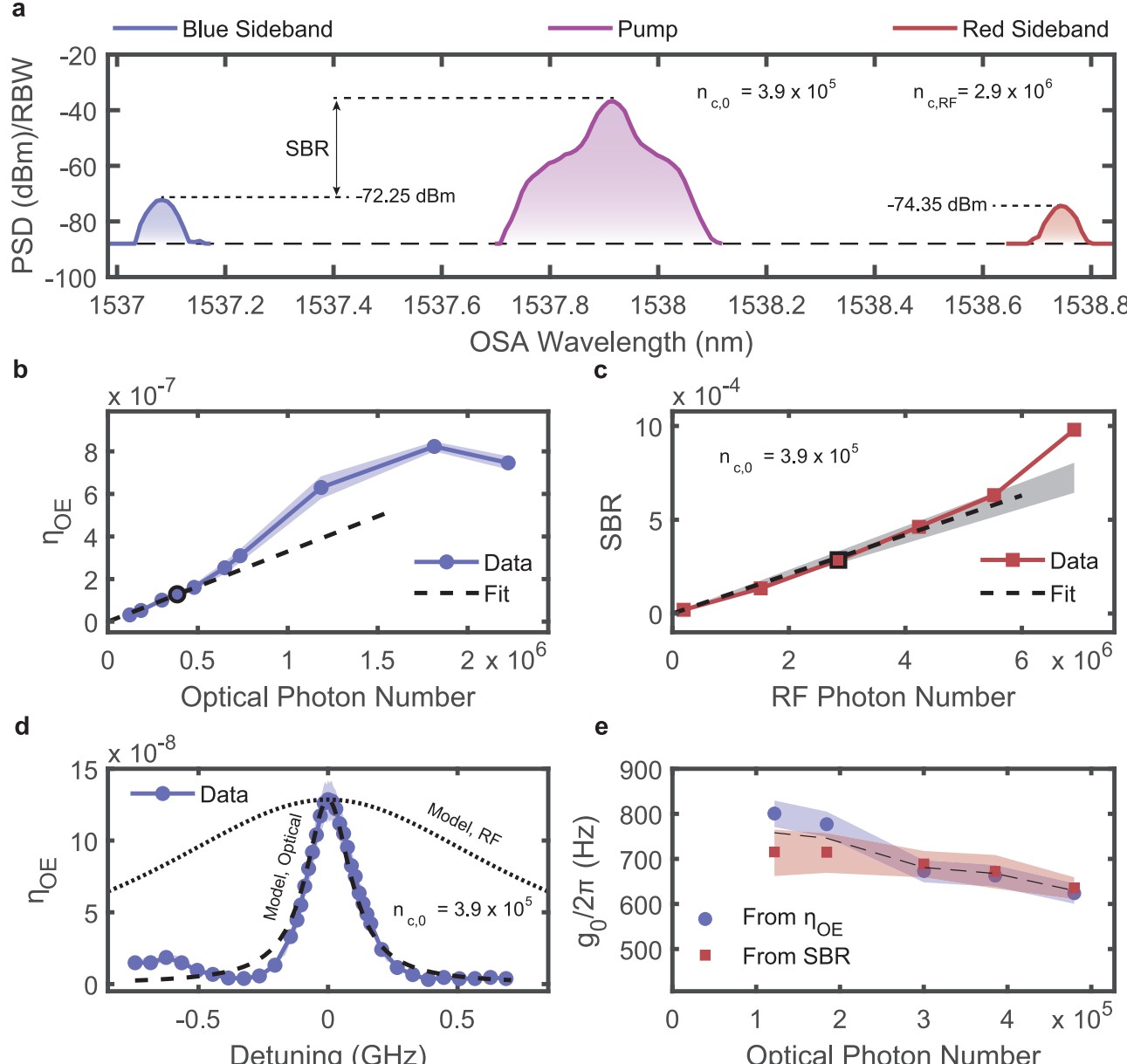

**Fig. 3 | Transducer characterization at T ≈ 4.9 K. a** Raw power spectral density data as measured from the OSA for modulation frequency $\Omega = 2\pi \times 105.285$ GHz. The input optical power in the TFLN waveguide is $P_{O,\text{inc}} = 46$ μW, and the incident mm-wave RF power is $P_{RF,\text{inc}} = 13$ μW; corresponding to intracavity photon numbers $n_{c,0} = 3.9 \times 10^5$ and $n_{c,RF} = 2.9 \times 10^6$, respectively. This results in a transduction efficiency, $\eta_{OE} = 1.3 \times 10^{-7}$, and a sideband ratio, $\Upsilon_{+,0} = 2.9 \times 10^{-4}$, shown as the highlighted blue circle and red square in subfigures (**b**, **c**), respectively. The resolution bandwidth of the OSA is 0.05 nm. **b** The transduction efficiency as a function of intracavity optical pump photon number, where the shading indicates 95% confidence interval. The dashed line depicts a linear fit to the low-power data, from which we can infer $g_0 \approx 2\pi \times 0.7$ kHz. **c** Side-band-ratio (SBR) versus intracavity mm-wave photon number. This is the ratio of the transmitted blue sideband power to the transmitted optical pump power. By fitting a line to the low-RF-power data

(dashed line), we can extract a separate measure of $g_0$, which agrees well with the results in subfigure (**b**). The gray shading indicates a 95% confidence interval for the linear fit. **d** Transduction efficiency as a function of modulation frequency ($\Omega$) for the highlighted blue circle in subfigure (**b**). The long-dashed Lorentzian is not determined by a fitting procedure, but instead by centering a Lorentzian function at the peak modulation frequency and setting its width to the independently measured blue mode optical linewidth. A similar Lorentzian model with the measured mm-wave linewidth is plotted as the short-dashed line. The shading indicates a 95% confidence interval. **e** Comparison of $g_0$ inferred from two analysis methods. Blue circles indicate values inferred from the loss-calibrated transduction efficiency (i.e., subfigure (**b**)). Red squares indicate values inferred from the self-calibrated SBR (i.e., subfigure (**c**)). Transparent lines show the 95% confidence interval at each optical power. The dashed line depicts the mean of the two curves.

where $T_{\text{eff}}$ is an effective temperature that encapsulates the combined effects of the equilibrium and non-equilibrium quasiparticle temperatures[40], $N(0)$ is the single-spin density of states at the Fermi level when the metal is in the non-superconducting state, and $\Delta(T_{\text{eff}})$ is the temperature-dependent superconducting gap from BCS theory[40].

To quantify the effects of the optically generated quasiparticles, we first determine the baseline temperature-dependent effects by

changing the cryostat's temperature and extracting the bare superconducting mm-wave cavity resonance frequency and internal loss rate, by fitting to our multiparameter model (see bottom panel of Fig. 2d), shown in Fig. 4a, b (left panel). With this baseline dataset, we can separately fit the mm-wave spectrum at each input optical pump power with the cryostat fixed at 4.9 K, plotted in Fig. 4a, b (right panel). The effective temperature is then defined by interpolating between the

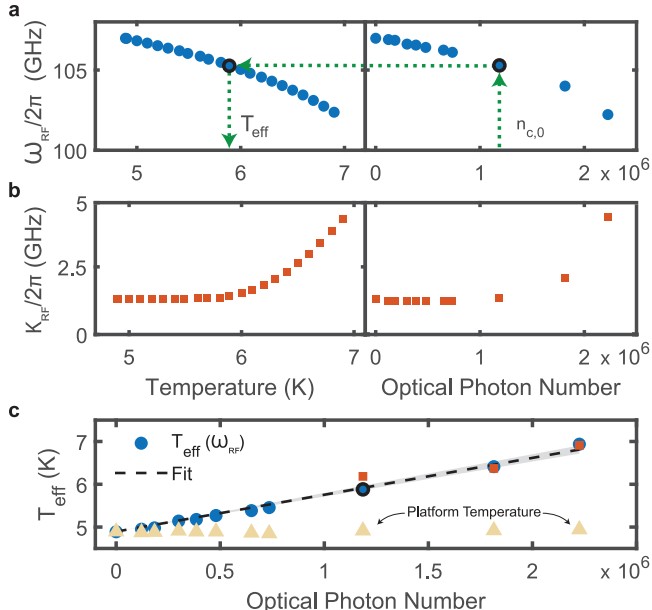

**Fig. 4 | Optically induced superconducting quasiparticle generation.**
**a** Millimeter-wave cavity frequency versus cryostat platform temperature (left panel) and versus intracavity optical pump photon number (right panel). Because optical photons generate quasiparticles, we can define an effective temperature following the green arrows: for each intracavity photon population ($n_{c,0}$), measure the mm-wave cavity frequency, find the corresponding frequency in the left panel (via interpolation), and report the corresponding temperature ($T_{eff}$). **b** Millimeter-wave cavity linewidth versus cryostat platform temperature (left panel) and versus intracavity optical pump photon number (right panel). **c** The effective temperature as a function of intracavity photon number: blue points are determined from the cavity frequency, and the red points are determined from the linewidth, using a similar procedure. The black-dashed line is a linear model, fit to the data, with the gray shading showing the 95% confidence interval. We determine the slope and corresponding 95% confidence interval to be 0.86 μK/photon (0.81, 0.90) μK/photon. Beige triangles indicate the cryostat platform temperature.

mm-wave resonance frequency shift due to the optical pump-induced quasiparticles and the equivalent resonance shift due to a change in cryostat temperature. This is depicted schematically by the green arrows in Fig. 4a. By performing this interpolation procedure for each optical power, we obtain a relationship between the effective temperature and the optical intracavity photon number, shown in Fig. 4c. Assuming that energy is only delivered to the superconducting cavity electrodes by scattered optical photons, we define a pump photon number-dependent effective temperature in our system by the following linear relationship,

$$T_{eff} = \frac{\partial T_{eff}}{\partial n_{c,0}} \cdot n_{c,0} + T_b, \tag{4}$$

where $\partial T_{eff}/\partial n_{c,0}$ is the effective temperature change per additional intracavity optical pump photon, $n_{c,0}$ is the optical intracavity photon number, and $T_b$ is the equilibrium bath temperature when there are no optical photons present in the cavity (equal to the cryostat platform temperature in this case). By fitting the data using Eq. 4, we obtain a value for $\partial T_{eff}/\partial n_{c,0}$, along with the 95% confidence interval, to be 0.86 μK/photon, (0.81, 0.90) μK/photon, with $T_b$ = 4.89 K.

## Discussion

In summary, we have demonstrated an integrated mm-wave cavity electro-optic transducer on TFLN. Our maximum observed efficiency is approximately 0.82 × 10⁻⁶ with a 3-dB bandwidth of $\kappa_+ \approx 2\pi \times 210$ MHz, closely following the linewidth of the optical mode.

Furthermore, we infer a single-photon coupling rate of $g_0 \approx 2\pi \times 0.7$ kHz. To understand and improve the performance of this device, we need to consider some key factors: fabrication, substrate mode mitigation, dispersion engineering, and operation. Note that being able to tune the photonic and/or the mm-wave resonances greatly improves ease of operation and fabrication yield, which can be addressed in tandem with the discussion below.

Our current fabrication process for the mm-wave cavity requires us to protect the TFLN racetrack by depositing oxide cladding atop the chip. However, we find that this oxide increases the loss of the mm-wave cavity. In addition to lowering the quality factor of the mm-wave cavity, the oxide cladding forces us to position the NbTiN electrodes farther away from the LN than necessary. This increased distance reduces the single-photon coupling rate, $g_0$, and thus the performance of the device. Therefore, we expect that eliminating the oxide cladding from our process would immediately improve the device. Some options for doing this would be to utilize lift-off instead of etching to define the NbTiN resonator or to use flip-chip methods to place the superconducting resonator closer to the optics.

The mm-wave substrate modes supported by the Sa substrate reduce the overall performance of our device. These substrate modes hybridize with the fundamental superconducting mode, thereby changing the electric field distribution. Consequently, the hybridization with the substrate modes impacts our device's performance in two ways: the field overlap between the mm-wave and optical modes reduces due to the increased mode volume of the hybridized mm-wave mode, reducing $g_0$; and the intracavity mm-wave photon number is modified by the introduction of additional detuning and broadening of the mm-wave cavity (see "Methods": "Substrate mode modeling and analysis" and "Methods": "Inferring the electro-optic coupling rate $g_0$"). Additionally, as seen in Fig. 2d and in Supplementary Fig. 5, these substrate modes complicate the fitting of the mm-wave spectrum. We envision two possible solutions to overcome the effects of substrate modes. First, by reducing the volume of the sapphire substrate (i.e., by changing the chip dimensions from 4.6 × 2.37 × 0.5 mm to 2.7 × 1.3 × 0.1 mm) we can reduce the number of substrate modes from 29 to 1 in the 95–110 GHz frequency range (see Supplementary Fig. 2). Secondly, through careful mm-wave engineering, it is possible to redesign the packaging to ensure that there is minimal coupling to substrate modes. Modifying the low-loss coupling structure in refs. 10,42 to accommodate TFLN photonics provides a platform that natively integrates with current mm-wave superconducting qubit packaging.

As observed in Fig. 3a, the present device generates both red- and blue-detuned optical sidebands due to having symmetric optical modes around the pump modes in the TFLN cavity. These sidebands correspond to simultaneous up and down conversion of the optical pump photons, which is undesirable for quantum applications, in which the second sideband acts as a parasitic process to the desired operation. For instance, when up-converting a mm-wave photon, the blue-detuned sideband corresponds to the desired signal, but the presence of the red-detuned optical mode allows for resonantly enhanced down conversion of the optical pump. That is, the optical pump photon combines with a mm-wave photon to down-convert into an entangled pair of a red-sideband optical photon and a mm-wave photon. In order to prevent this back-conversion effect, we must suppress one of the two optical sidebands. We could achieve this with a dispersive element to selectively shift one of the three optical modes. In doing so, we modify the TFLN cavity spectrum to feature pairs of modes, ensuring that only one sideband is resonantly enhanced, while the other is suppressed[16,17,34].

Improving the device's thermal operation is critical for quantum applications. All of the experiments discussed in this work are carried out with a continuous-wave optical pump, which effectively bathes the superconducting electrodes in a constant stream of scattered photons.

Absorption of these photons is analogous to increasing the temperature of the electrodes, as shown in Fig. 4. The absorbed optical photons break Cooper pairs and generate non-equilibrium quasiparticles in the superconducting electrodes. While the frequency of the superconducting resonance decreases, the linewidth of the resonance increases, with both effects deteriorating the device performance (see Fig. 3b). A common technique used for microwave-to-optical transducers is to pulse the optical pump. By pulsing the pump, time is provided for the quasiparticle density to relax to its thermal equilibrium value. This enables ideal superconducting conditions for each transduction event (mediated by the pump-pulse).

Furthermore, in this work, we operate the device at roughly 5 K. At this temperature, the average thermal photon occupation is approximately 0.6 per mode at 105 GHz. To perform any quantum experiment, we would need to operate at a lower temperature. For the same frequency, operating at 1 K, the thermal occupation is less than 0.01 photons per mode, which is sufficiently low[10]. Operating at a lower temperature comes with the additional benefit that the superconducting mm-wave resonator will have an increased quality factor. Here, however, the 5 K operating point serves as a proof of demonstration that such transducers could operate at elevated temperatures in a dilution refrigerator, thereby leveraging the greater cooling power of higher-temperature stages[8,10].

Overall, we have implemented a mm-wave cavity electro-optic system and demonstrated coherent mm-wave-to-optical transduction. Our work shows that integrated devices operating at mm-wave frequencies are not only feasible but also potentially advantageous for future quantum applications by enabling higher-temperature operation. With a clear path forward to improve our device, we envision a new regime of superconducting quantum technologies operating at mm-wave frequencies. Our work ultimately highlights the challenges inherent to designing integrated mm-wave photonic systems and provides a path toward future advancements in mm-wave quantum hardware.

## Methods
### Experimental setup
The complete experimental setup, depicted in Supplementary Fig. 6, can be separated into two optical and two electronic signal paths. These paths comprise: the primary optical path for optical sideband measurements; a secondary path for self-heterodyne measurements (Fig. 2b, c); the primary mm-wave RF path; and the electronic signal detection path, used to lock the optical pump to the mode $\omega_0$. Please see Supplementary Information S4 for detailed descriptions of these different paths. Supplementary Fig. 11c presents an optical image of the inside of the cryostat, showing the WR10 waveguides and part of the mm-wave waveguide network.

### Pump detuning calibration and locking
We lock our laser to the optical cavity using a Red Pitaya in combination with open source software, PyRPL. We control our laser's (Santec TSL-710) wavelength via voltage applied to a piezoelectric actuator inside the laser, which is set by either the Red Pitaya output or from our DAQ's analog output directly, selected using a mechanical BNC relay. Using PyRPL, we specify a voltage corresponding to transmission slightly detuned from the resonance of the optical cavity pump mode and monitor the output transmission as a voltage on our APD. This voltage is routed to the Red Pitaya, and the PID loop varies the piezo voltage sent to the Santec to maintain a constant transmission voltage. We tune into the PID lock point from the blue side of the mode, which we observe is a more stable operating point than the red-detuned side. Upon initially locking to the mode, the optical cavity shifts dramatically as a result of the thermo-optic and photorefractive effects in TFLN[43]. We find that after pumping the cavity continuously for some time, the rate of cavity drift drops significantly, which we attribute to

charge carriers saturating trap sites until we reach a steady state (see Supplementary Information S5 for more details).

The exact detuning of the locked pump from the optical resonance ($\Delta_0$) is critical to our analysis, but direct measurement is difficult. This value is obtained through a single parameter fit of the transmission $T = |1 - \frac{\kappa_{e,0}}{i\Delta_0 + \kappa_0/2}|^2 = P_{OSA}/P_{PM} \cdot \eta_0$, where $T$ is the transmission through the optical pump cavity, $P_{OSA}$ is the measured optical pump power on the OSA at the pump wavelength, $P_{PM}$ is the optical power reading from a power meter right before the laser is sent into the cryostat, and $\eta_0$ is a measured factor, which encapsulates measurements of path losses, beam-splitter ratios, grating coupler efficiency, and differences in calibration between the OSA and PM. We infer $\kappa_{e,0}$ and $\kappa_0$ from self-heterodyne measurements, as described in the main text (more details are provided in the Supplementary Information Section S2.2).

### Device fabrication and packaging
Our fabrication closely follows that of McKenna and Witmer et al.[16]. We start with a $12 \times 16$ mm piece of material with nominally 500 nm of MgO-doped $X$-cut LN atop a 500 μm-thick $C$-cut Sa. The optical waveguides, the racetrack resonator, and the grating couplers are defined using hydrogen silsesquioxane (HSQ) resist and electron beam lithography (JEOL, JBX-6300 FS, 100 keV). Using an Intlvac Ion Beam Mill (argon ions), we etch roughly 300 nm of TFLN to transfer the mask into the TFLN.

Next, we etch excess TFLN around the optical components. We define a second mask using SPR3612 resist and photolithography (Heidelberg MLA 150, 375 nm) and etch the remaining roughly 200 nm of TFLN slab around the optical devices via the Intlvac ion mill. At this stage, we clean the sample using various acids, followed by annealing at 500 °C in an atmosphere.

We next clad the chip with approximately 2 μm of silicon dioxide (SiO$_2$) using low-temperature high-density chemical vapor deposition (PlasmaTherm Versaline HDP-CVD), to protect the optical components from NbTiN deposition in a later step. Cladding takes two steps of roughly 1 μm depositions, separated by a 530 °C anneal in atmosphere[44]. As discussed in the main text, SiO$_2$ contributes to the loss in the mm-wave resonance, so we etch windows into the oxide where we intend to place the mm-wave cavity. Using the SPR220-3 resist, we pattern windows via another round of photolithography and etch the SiO$_2$ with fluorine chemistry in an inductively coupled plasma reactive ion etcher (PlasmaTherm ICP RIE). With the Sa exposed in these regions, we deposit NbTiN via DC magnetron co-sputtering of niobium (Nb) and titanium (Ti) in a mixed nitrogen-argon environment (Kurt J. Lesker PVD Pro-line).

In a third round of photolithography, we pattern the mm-wave cavities with SPR3612 resist. Lastly, we etch the NbTiN with a mixture of SF$_6$/Ar (PlasmaTherm ICP RIE). This device is pictured in Supplementary Fig. 11a, b.

### Superconducting millimeter-wave cavity design
We design the mm-wave RF cavity such that it maintains a predominantly single-polarity voltage drop across the TFLN. Due to RF loss in the SiO$_2$ cladding, we cannot cross the oxide with the resonator and therefore carefully design a half-wave-like resonator. We modify a half-wave transmission coplanar stripline resonator so that the polarity of the electric field does not change sign between the electrodes in the TFLN region (see Supplementary Information Section S1.2.1 for additional details). We present the current density distribution and the electric field distribution of the mode in Supplementary Fig. 1a–c.

The dimensions of the mm-wave device dictate its resonant frequency given a fixed kinetic inductance. Predominantly, the resonator length ($L$) determines its fundamental frequency, whereas the length of the coupling section ($L_c$) and size of the capacitive pads control the external coupling rate of the resonator to the WR10 waveguide

(labeled in Supplementary Fig. 1a). We model the fundamental frequency as $\omega_{RF}(L) = v_p/L + \delta\omega$, where $v_p$ is the phase velocity of the coplanar stripline section, and $\delta\omega$ describes a shift in resonance frequency due to the capacitive coupling region (the pads and coupling section).

To incorporate the effects of kinetic inductance in the design, we estimate the sheet inductance (in units of inductance per square) using the relationship $L_s \approx \hbar/(\pi\Delta_0) \times R_s$. In this expression, $L_s$ is the kinetic sheet inductance, $\Delta_0 \approx 1.764 k_B T_c$ describes the superconducting gap at $T = 0$ K of the material, and $R_s$ describes the film's normal-state sheet resistance (in units of resistance per square). During fabrication of the main device, we include a witness sample so that we can measure the film sheet resistance and thickness, and infer the resistivity. For the device presented in this paper, we measure $\rho_s \approx 250\ \mu\Omega$ cm for a 50 nm film on the witness chip. From here, we can estimate $L_s \approx 6.9$ pH/□ (pH/ "square") assuming $T_c \approx 10$ K. With this estimate of the sheet inductance, we design superconducting test devices (using SONNET Software[45]) and measure them before continuing fabrication on the main device. We summarize the measurements of the NbTiN-witness devices against our simulation of the expected resonance frequencies in Supplementary Fig. 1d. The relationship between the measured resonance and simulated resonance frequencies can be shown to be linear,

$$\omega_{RF}^{actual} = r\omega_{RF}^{sim} + \epsilon. \tag{5}$$

Using this model, we can design and pattern the NbTiN resonators on the transducer device. We sweep the design parameters of the mm-wave cavity across the fabricated devices to increase robustness against film and optical device variations. Based on the measured optical and mm-wave characteristics of the various fabricated devices, we select one for detailed study in this manuscript. Note that after incorporating real device dimensions (as measured from scanning electron microscopy), we use SONNET Software[45] to infer the sheet inductance to be $L_s^{meas} \approx 7.6$ pH/□, by matching the simulated resonant frequency to the measured one; this sheet inductance value is within 10% of our estimate based on the BCS theory.

## Substrate mode modeling and analysis

The approximately 500 μm-thick substrate, which is a standard thickness for Lithium Niobate-on-Sapphire (LiSa) wafers, acts as a cavity that supports many mm-wave substrate modes. To quantify these substrate modes, we perform eigenmode simulations in COMSOL[46] and count the number of modes between 95 and 110 GHz, as a function of geometry. In Supplementary Fig. 2a, we show the geometry of our simulation, where all boundaries are perfect electrical conductors except the waveguide input and output ports. As the thickness of the sapphire decreases, the number of supported modes also decreases. Some, but not all, of these substrate modes are visible in the mm-wave cavity spectrum (see Fig. 2e). The frequency domain simulation in Supplementary Fig. 2d exhibits only a handful of modes, although we predict 29 eigenmodes to be supported. This suggests that only modes with polarization similar to the TE10 mode of the WR10 waveguide are driven.

Therefore, in our model, we assume that the substrate modes are not directly accessible or driven via the WR10 waveguide field. Only substrate modes with an appropriately matched polarization and spatial mode profile can be coupled to the superconducting resonator and are visible in the measured mm-wave RF spectrum (see Fig. 2e). This coupling, which hybridizes the superconducting mm-wave mode with the substrate modes, leeches mm-wave photons from the superconducting cavity. The result of this hybridization is an effective loss increase and detuning of the superconducting mm-wave cavity.

The mm-wave spectrum, accounting for these hybridized substrate modes, is then described by:

$$S_{21}^{RF} = 1 - \frac{\kappa_{e,RF}/2}{i(\Delta_{RF} + \widetilde{\delta}) + (\kappa_{RF} + \widetilde{\gamma})/2} \tag{6}$$

From Eq. 6, we see the overall effect of the substrate modes is to increase loss and detuning described by:

$$\widetilde{\delta} = -\sum_{n=1}^{N} \frac{|J_n|^2 \Delta_n}{\Delta_n^2 + \gamma_n^2/4} \tag{7}$$

$$\widetilde{\gamma}/2 = \sum_{n=1}^{N} \frac{|J_n|^2 \gamma_n/2}{\Delta_n^2 + \gamma_n^2/4}, \tag{8}$$

where $J_n$ is the coupling rate, $\Delta_n = \omega_n - \Omega$ is the detuning and resonant frequency, and $\gamma_n$ is the total linewidth of the $n$th substrate mode. We fit the complete multiparameter mm-wave transmission, $S_{21}^{RF}$, using an iterative procedure with particle swarm optimization (PSO) to obtain the substrate mode parameters. These parameters are given in Table 1.

In our modeling, we include eight unique substrate modes that hybridize with the mm-wave cavity. We determine the number of hybridized substrate modes through a careful fitting and analysis procedure detailed in Supplementary Information S3. The procedure ensures that each included substrate mode can be traced to an avoided crossing that appears in the data (see Fig. 2d, e). To provide some intuition, we show in Supplementary Fig. 5a the impact of consecutively including the substrate modes one-by-one at 4.9 K. Additionally, we highlight the accuracy of our model (see Eqs. 6–8) for temperatures between 4.9 and 6.9 K by plotting the theoretical prediction and the data in Supplementary Fig. 5b.

## Inferring the electro-optic coupling rate $g_0$

We infer the zero-point electro-optic coupling rate using two separate analyses. The first approach uses the measured transduction efficiency, and the second uses the measured optical sideband ratio. Additionally, as discussed in the previous section, coupling to substrate modes eventually impacts the mm-wave intracavity population by adding an additional shift to the mm-wave detuning and total linewidth. Thus, the on-chip efficiency of the transduction process and the sideband ratio are modified (when compared with input-output theory),

$$\eta_{OE} \approx g_0^2 \cdot \left( \frac{\kappa_{e,+}}{\Delta_+^2 + \kappa_+^2/4} \right) \cdot \left( \frac{\kappa_{e,RF}/2}{(\Delta_{RF} + \widetilde{\delta})^2 + (\kappa_{RF} + \widetilde{\gamma})^2/4} \right) \cdot n_{c,0}, \tag{9}$$

$$\Upsilon_{+,0} \approx g_0^2 \cdot \left( \frac{\omega_p + \Omega}{\omega_p} \right) \cdot \left( \frac{\kappa_{e,+}}{\Delta_+^2 + \kappa_+^2/4} \right) \cdot \left( \frac{\kappa_{e,0}/(\Delta_0^2 + \kappa_0^2/4)}{1 - 4\kappa_{e,0}(\kappa_0 - \kappa_{e,0})/(4\Delta_0^2 + \kappa_0^2)} \right) \cdot n_{c,RF}, \tag{10}$$

where, $n_{c,RF} = (\kappa_{e,RF}/2)/((\Delta_{RF} + \widetilde{\delta})^2 + (\kappa_{RF} + \widetilde{\gamma})^2/4) \cdot \frac{P_{RF}}{\hbar\Omega}$ and $n_{c,0} = \kappa_{e,0}/(\Delta_0^2 + \kappa_0^2/4) \cdot \frac{P_0}{\hbar\omega_p}$. Notice that deducing the electro-optic coupling rate from transduction efficiency is sensitive to both the optical and RF path losses and efficiencies, whereas the sideband ratio is sensitive only to RF path loss and efficiencies because the optical path loss normalizes out in this case, as discussed in the main text.

To take stock of all measurements, we provide a brief description of how each value is determined in Eqs. 9 and 10 (see Supplementary Information S2 for a more detailed discussion of this procedure).

Firstly, all optical $\kappa$'s and $\kappa_e$'s are inferred through the self-heterodyne measurement described in the main text. The optical pump detuning $\Delta_0$ is measured as described in "Methods": "Pump detuning calibration and locking." The blue sideband detuning can be written as, $\Delta_+ = \mathrm{FSR}_+ + \Delta_0 + \Omega$. We write it in this form because in our optical spectroscopy, differences in frequency are more accurately calibrated than the absolute optical wavelength. Note that the quantity $\mathrm{FSR}_+$ is the frequency difference between the pump mode and the blue sideband mode, obtained via the self-heterodyne measurement described above. Lastly, the optical path losses and photodetector power are measured so that we can infer the on-chip photon flux, and thus compute $n_{c,0}$, assuming the input and output grating coupler losses are equal.

For all mm-wave modal parameters, $\Delta_{\mathrm{RF}}, \widetilde{\delta}, \kappa_{\mathrm{RF}}, \widetilde{\gamma}$, and $\kappa_{e,\mathrm{RF}}$ are determined from the multiparameter fit, to include the effects of the hybridized substrate modes. To determine the RF path loss, we perform a cryogenic calibration procedure (see Supplementary Information S2.1)[8], which tells us the incident mm-wave RF power on the chip and thus provides an estimate of $n_{c,\mathrm{RF}}$ required for Eq. 10.

As stated in the main text, we observe that the estimate for $g_0$ from Eqs. 9 and 10 converges, providing evidence that the path calibrations and estimates for the mode parameters are self-consistent.

## Data availability
All data used in this study are available from the corresponding authors upon request.

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

## Acknowledgements

The authors acknowledge Wentao Jiang and Felix Mayor for their assistance in Red Pitaya setup and fiber gluing, Hubert Stokowski and Timothy McKenna for useful discussions on electro-optic theory and modeling, Sandesh Kalantre, Devin Dean, Matthew Maksymowych, Erik Szakiel, Luke Qi, and Samuel Gyger for their help in NbTiN sputtering, and Monika Schleier-Smith and Paul Welander for their experimental support. Lastly, the authors acknowledge Debadri Das and Samuel Gyger for insightful discussions. This work was supported by the U.S. Army Research Office (ARO)/Laboratory for Physical Sciences (LPS) Modular Quantum Gates (ModQ) program (Grant No. W911NF-23-1-0254), by the Air Force Office of Scientific Research and the Office of Naval Research (Grant No. FA9550-23-1-0338), the US Department of Energy through (Grant No. DE-AC02-76SF00515) and via the Q-NEXT Center. This work was also funded by Amazon Web Services. K.K.S.M. gratefully acknowledges support from the Natural Sciences and Engineering Research Council of Canada (NSERC). J.F.H. gratefully acknowledges support from the NSF GRFP (Grant No. DGE-1656518). Part of this work was performed at the Stanford Nano Shared Facilities (SNSF) and Stanford Nanofabrication Facility (SNF), supported by the National Science Foundation (Grant No. ECCS-2026822).

## Author contributions

K.K.S.M. and J.F.H. jointly led the work, conceived the experiment, fabricated, and measured the devices. A.H.S.-N. provided additional theoretical and experimental support. E.N. and A.H.S.-N. supervised all efforts. K.K.S.M. and J.F.H. wrote the manuscript with input from all authors.

## Competing interests

A.H.S.-N. is an Amazon Scholar. The other authors declare no competing interests.
