## [Transparent Peer Review file · Nature Communications]

Integrated millimeter-wave cavity electro-optic transduction

Corresponding Author: Mr Kevin Multani

Version 0:

Reviewer comments:

Reviewer #1

(Remarks to the Author)

The manuscript titled "Integrated sub-terahertz cavity electro-optic transduction" demonstrate 100 GHz cavity-enhanced electro-optic transduction leveraging optical modes and a superconducting resonator. Beyond the previous developments in microwave-to-optical quantum transduction, this work extends the frequency to THz range and showing promising performance. Overall, I believe this work renders a significant advance and will be interesting to broad audience of nature communications. Below are some technical comments, I hope authors find them useful.

-- I notice that in this work uses only a single optical cavity and uses optical modes in the same family separated by FSR. Thus, the THz signal will drive the pump light to both the blue and red side modes, and potential to higher order of blue/red side modes as well. Will that be any issue or disadvantage of this configuration? If I remembered correctly most previous work use coupled ring resonators, then the transduction only limit to two involved modes. Could authors discuss and provide some insights?

-- what's the possible THz power delivered to the chip? Is that strong enough to see Autler Townes splitting or AC stack shift in the optical transmission spectra? That could enable more interesting regime for more fruitful physics.

Below are some detailed technical questions from my Early Career co-Reviewer, I would like to pass to authors as is. I believe those questions bring some interesting points from Early Career researchers.

-- In line 128, the authors mentioned sub-THz mode red-shift and broaden due to thermal excitation. However, the superconducting mode broadening due to temperature change is not very significant from Fig.2 d. Is there any more intuitive way to show this trend?

-- In Fig.1 c, what is the function of bottommost sapphire region since it does not intersect with the EO interaction region?

-- In extended Fig.1, does the left VNA provide the sub-THz pump signal when the device is working in transduction operation? And how is the WR10 waveguide connected to the electrodes in 4.9 K cryogenic environment?

-- In line 266, the authors mentioned device thermal occupation should be lower at 1 K. Why did the authors not use the dilution fridge to achieve a deeper cryogenic environment to investigate the device performance?

Reviewer #2

(Remarks to the Author)

Reviewer #3

(Remarks to the Author)

In "Integrated sub-terahertz cavity electro-optic transduction," the authors present an electro-optic (EO) transducer based on

thin-film lithium niobate (TFLN) on sapphire that converts sub-THz signals to telecom optical signals. The device employs a sub-THz NbTiN resonator positioned adjacent to the TFLN to modulate the optical signal that passes through the TFLN waveguide that is buried beneath silicon dioxide. While the idea of interfacing sub-THz and optical signals using an integrated EO device is interesting, significant gaps remain in both the data analysis and explanations, leaving key questions unanswered. Furthermore, the organization of the paper appears unpolished; the authors frequently direct readers to the Supplementary Information without specifying which particular sections contain the relevant details.

In conclusion, the manuscript suffers from imprecise language throughout. The vague and incomplete explanations compromise the clarity and rigor expected for publication. Based on these issues in presentation and analysis, I do not recommend publication of this work in Nature Communications.

Below are outlined questions and comments to the authors.

1. The authors mentioned the existence of parasitic modes in the system, but they provide little analysis and explanation of this parasitic mode. What is the exact origin of these modes, and how do they affect the device performance?
2. The aforementioned “parasitic modes” later become the “substrate mode” in the writing, yet there’s little explanation provided in the main text about it. What’s the difference between a parasitic mode and a substrate mode?
3. The authors should specify the exact section in the Method and Supplementary Information that readers should consult, rather than pointing to the entire document.
4. The data and fitting in Fig. 2e are not convincing. With multiple peaks/dips in the data, the “multi-parameter model” is not reliable. “The substrate modes by transparent red vertical lines” hardly match the peaks.
5. In Fig. 2d, why do the modes’ frequencies not drift continuously with temperature? Rather, they seem to “hop” from one vertical straight line to another. The amplitude of a certain line also changes non-monotonically. These need explanation.
6. How does the author estimate the coupling rate accurately in the presence of the substrate modes? Is it possible to filter out the substrate mode response? And would this unwanted hybridization to the substrate mode limit the theoretical transduction efficiency?
7. In the heterodyne measurements used in Fig. 2, the authors stated that these measurements could lead to the determination of total coupling rate and loss rate. However, no further information about this is provided. How to infer the coupling rate using the heterodyne measurement result?
8. In the transducer operation section, the term “laser piezo voltage” needs clarification. Does this refer to a piezoelectric actuator that controls the laser frequency, or does it serve another function in the measurement?
9. The author also mentioned an asymmetry of the coupling rate to the red and blue sidebands in Fig. 2a. But in Fig.2a, the red and blue sidebands seem similar in their signal intensity without a clear asymmetry. The author should clarify what asymmetry they are referring to. In addition, the author should also indicate the supporting information section that the reader should read.
10. The authors attribute the non-linear response in the transduction efficiency vs pump optical photon (Fig. 3b) to the local heating of the sub-THz resonator even at the low-cooperativity limit. This raises a fundamental concern: if heating effects are already significant at the current efficiency levels, how can this device achieve high transduction efficiency when strong optical pumping is required for optimal performance? Furthermore, this device is fabricated on sapphire, which intuitively should provide a better heat dissipation channel; however, the heating effect is surprising and warrants further explanation. The author should provide a more detailed analysis of this.
11. What is the typical quality factor (Q) of the sub-THz cavity? In Fig. 4, the Q seems to be $\sim 107/2 = 53.5$ at low optical pumping. Is this considered a high-Q sub-THz resonator? How about the optical Q of the TFLN racetrack cavity? The triply resonant process should benefit from a high Q from both the sub-THz cavity and the optical cavity.
12. In Figure 4, the authors present the linewidth and frequency shift due to the optical heating. The author uses an effective temperature to calculate the local heating of the device. What is the definition of the effective temperature? In addition to a rise in temperature near the electrode, how about the added noise to the system?

Reviewer #4

(Remarks to the Author)

I co-reviewed this manuscript with one of the reviewers who provided the listed reports. This is part of the Nature Communications initiative to facilitate training in peer review and to provide appropriate recognition for Early Career

Researchers who co-review manuscripts.

Reviewer #5

(Remarks to the Author)

This manuscript presents the experimental demonstration of a triply-resonant cavity electro-optic transducer operating at ~100 GHz microwave range, implemented via integration of a superconducting NbTiN sub-THz resonator and a TFLN optical racetrack resonator. The authors report a single-photon electro-optic coupling rate of 0.7 kHz and a transduction efficiency 0.82×10^{-6} . The work addresses important engineering and physical challenges in realizing microwave-to-optical transduction at relative high frequency (~100 GHz), and offers thorough characterization. Electro-optic transduction at millimeter wave range is crucial for many applications. Achieving direct transduction at such a high microwave frequency typically encounters lots of key challenges. Therefore this reviewer do think this work represents a significant step and recommends the publication of Nature Communications. Below are some comments that may help improve the manuscript:

1. I think using millimeter wave is probably more suitable than sub-terahertz. This work demonstrates transduction at ~100GHz microwave frequencies, which exactly hits the range of millimeter wave and is a bit far from terahertz.
2. The authors here used a single cavity with different modes separated by the FSR. It seems the drawback would be that both up and down conversion will happen at this time? The authors may want to comment on this a bit in the manuscript.
3. Following up on comments 2, in Figure 3a, one could also see that both up and down sidebands show up. Then it might be a bit confusing for the readers that the authors only draw the up-conversion process in the Figure 1a.

Version 1:

Reviewer comments:

Reviewer #1

(Remarks to the Author)

Both myself and my co-reviewer believe our previous concerns have been fully addressed. We recommend this manuscript for publication.

Reviewer #2

(Remarks to the Author)

Reviewer #3

(Remarks to the Author)

The authors' responses to my comments are satisfactory. I'm glad to see the revised manuscript has improved considerably from the original, so I support its publication.

Reviewer #4

(Remarks to the Author)

Reviewer #5

(Remarks to the Author)

The authors have addressed my comments and I recommend the publication.

Reviewer #1 (Remarks to the Author):

The manuscript titled “Integrated sub-terahertz cavity electro-optic transduction” demonstrate 100 GHz cavity-enhanced electro-optic transduction leveraging optical modes and a superconducting resonator. Beyond the previous developments in microwave-to-optical quantum transduction, this work extends the frequency to THz range and showing promising performance. Overall, I believe this work renders a significant advance and will be interesting to broad audience of nature communications. Below are some technical comments, I hope authors find them useful.

We thank Reviewer 1 for their comments, questions, and positive outlook on our work. Please find below our point-by-point responses.

-- I notice that in this work uses only a single optical cavity and uses optical modes in the same family separated by FSR. Thus, the THz signal will drive the pump light to both the blue and red side modes, and potential to higher order of blue/red side modes as well. Will that be any issue or disadvantage of this configuration? If I remembered correctly most previous work use coupled ring resonators, then the transduction only limit to two involved modes. Could authors discuss and provide some insights?

Response. Reviewer 1 is correct to point out the dual sideband generation. In our original submission, we briefly commented on the impact of the red-detuned optical mode in lines 81-86. In particular, the blue-detuned optical mode enables a beam-splitter-type interaction in the system, (the results of which are detailed in our work). However, the red-detuned optical mode enables an entangled pair generation process, in which an optical pump photon splits into an entangled pair of a microwave photon and a down-shifted optical photon. In our system, both of these processes happen simultaneously, which hinders operation as a single-photon quantum transducer. In many quantum transduction schemes, only *one* of these two processes is desired. A common approach to isolate these processes is to use a pair of coupled cavities. The coupling between the two cavities locally modifies the optical mode spectrum so that only a single pair of modes are modulated by the applied RF power. In the present work, the red sideband has no impact on device characterization i.e. efficiency (η) and interaction rate (g_0) measurements, because we do not investigate these two quantum-mechanical processes at a single-photon level.

Actions taken. In order to clarify these points, we have added lines 301-312 in the manuscript’s discussion section.

-- what’s the possible THz power delivered to the chip? Is that strong enough to see Autler Townes splitting or AC stack shift in the optical transmission spectra? That could enable more interesting regime for more fruitful physics.

Response. With the current device, observing Autler-Townes splitting is not possible. The highest rf power we apply in this work is roughly 32 μ W incident to the device. Our rf source

outputs about 1.9 mW at 105 GHz with an rf path propagation efficiency of approximately 1.7%. To observe optical mode splitting, the total rf-optical photon interaction rate, $G/2\pi$ (where $G = \sqrt{n_{\text{RF}}} g_0$), must be comparable to the optical mode linewidth, 210 MHz. The required number of mm-wave photons in the cavity is given by $n_{\text{RF}} = (G/g_0)^2 = (210 \text{ MHz}/700 \text{ Hz})^2 = 90e9$, corresponding to an incident mm-wave power $P_{\text{inc}} = 270 \text{ mW}$ with our current device parameters. Unfortunately, the currents within the superconducting electrodes would be much greater than the critical current at this power level. Additionally, one would need about 16 W of mm-wave power to deliver 270 mW to the chip, given a path efficiency of 1.7% (this is a non-trivial power level at these frequencies). This could of course be improved by reducing the optical mode linewidth (state of the art lithium niobate cavities can exhibit linewidths of a few tens of MHz) and improving the electric field overlap.

However, we also agree with Reviewer 1 that this system could enable interesting physics. For most demonstrations of microwave-to-optical transduction implemented via optomechanics or electro-optics, the conversion bandwidth is limited by the microwave linewidth, which is typically much narrower than the optical mode linewidth. In our system the opposite is true: our rf linewidth is greater than our optical linewidth. This puts us in a so-called “reversed-dissipation regime” (lines 198-201 in the original manuscript). In this regime, when the electro-optic cooperativity approaches unity, $C \sim 1$, electro-optic induced amplification or cooling of an optical mode could be studied.

Action taken. In the revised manuscript we include some additional context regarding the requirements of the reversed-dissipation regime for observing amplification/cooling, and we note the maximum cooperativity measured in this work, $C \sim 1e-5$ (see lines 231-236 in the revised manuscript).

Below are some detailed technical questions from my Early Career co-Reviewer, I would like to pass to authors as is. I believe those questions bring some interesting points from Early Career researchers.

We appreciate the Early Career co-Reviewers’ time and thoughtful feedback.

-- In line 128, the authors mentioned sub-THz mode red-shift and broaden due to thermal excitation. However, the superconducting mode broadening due to temperature change is not very significant from Fig.2 d. Is there any more intuitive way to show this trend?

Response. While in the original submission we demonstrate this effect in Fig. 4a, we acknowledge that the language used in the text and in Figs. 2d,e is not clear. The reasoning why it is difficult to see the broadening of the rf mode is two-fold:

1. the hybridization between the rf cavity mode and the chip substrate modes leads to avoided crossings that make the spectrum look fragmented.

2. if the cavity mode were not hybridized with the substrate modes, the lineshape would lose contrast as it became broader.

These comments are similar to many of the comments from Reviewers 3. Please see further details in our responses to their comments below.

Actions taken. As this comment overlaps with the comments received from Reviewer 3, we provide a detailed account of revisions in our responses to them. Please see details in our responses below, and note the changes to text and figures in our re-submitted redlined manuscript. In summary of the applied changes, we revised Figs. 2d,e, Fig. 4, and included major revisions to the text, especially in the section titled “Optical & millimeter-wave cavity spectroscopy”.

-- In Fig.1 c, what is the function of the bottommost sapphire region since it does not intersect with the EO interaction region?

Response. The main reason for the bottom-most etched-window is to keep the symmetry between the top and bottom straight sections of the racetrack resonator. The device would still function if that bottom-most window was not etched.

Action taken. N/A.

-- In extended Fig.1, does the left VNA provide the sub-THz pump signal when the device is working in transduction operation? And how is the WR10 waveguide connected to the electrodes in 4.9 K cryogenic environment?

Response. Yes, the VNA provides the mm-wave pump signal. The VNA has hardware options that enable compatibility with frequency extension modules (RFE in Extended Fig.1). These modules use the VNA's internal IF and LO to up-mix and down-mix rf signals in the W-band (75 GHz-110 GHz). In transducer operation, we only need to supply a constant RF frequency, which we do by setting the VNA's output to CW (continuous wave) mode, instead of frequency sweep mode. We are also able to set this CW frequency, which is how we perform measurements of the transducer bandwidth (Fig. 3d). Separately, we perform rf spectroscopy of the superconducting cavity using the frequency sweep mode.

In our design, the electrodes of the superconducting cavity act as an antenna. The device is mounted in a copper holder (shown in Fig. 1), which is clamped between two WR10 waveguides. To get a better sense of the geometry, Extended Figs. 4a,b show images of the copper holder with the packaged chip.

Action taken. N/A

-- In line 266, the authors mentioned device thermal occupation should be lower at 1 K. Why did the authors not use the dilution fridge to achieve a deeper cryogenic environment to investigate the device performance?

Response. We acknowledge the merits of performing these measurements in a deeper cryogenic environment, namely a lower thermal occupation and a potentially higher superconducting cavity quality factor. However, low thermal occupation is only critical for single-photon operation of the transducer. In our work, we perform a proof-of-concept experiment characterizing a fully-resonant modulator in the coherent (i.e., “classical”) regime, so high thermal occupation is irrelevant to our measurements/results. Integrating mm-wave rf sources and signals within the physical environment/structure of a dilution refrigerator is also very challenging. For the purposes of this work, we did not believe that our demonstrations warranted the high overhead required to implement these devices in a dilution refrigerator environment. In fact, we believe demonstrating use in a closed-cycle 4 K cryostat emphasizes the potential benefits of developing these transducers at mm-wave frequencies.

Reviewer #2 (Remarks to the Author):

Reviewer #3 (Remarks to the Author):

In “Integrated sub-terahertz cavity electro-optic transduction,” the authors present an electro-optic (EO) transducer based on thin-film lithium niobate (TFLN) on sapphire that converts sub-THz signals to telecom optical signals. The device employs a sub-THz NbTiN resonator positioned adjacent to the TFLN to modulate the optical signal that passes through the TFLN waveguide that is buried beneath silicon dioxide. While the idea of interfacing sub-THz and optical signals using an integrated EO device is interesting, significant gaps remain in both the data analysis and explanations, leaving key questions unanswered. Furthermore, the organization of the paper appears unpolished; the authors frequently direct readers to the Supplementary Information without specifying which particular sections contain the relevant details.

In conclusion, the manuscript suffers from imprecise language throughout. The vague and incomplete explanations compromise the clarity and rigor expected for publication. Based on these issues in presentation and analysis, I do not recommend publication of this work in Nature Communications.

We thank the reviewers for their thorough and critical reading of the manuscript. While we believe that many of the reviewers’ questions are addressed in the original manuscript, we have clarified and presented more clearly to mitigate confusion. We have substantially revised the manuscript to address these concerns, as addressed point-by-point below.

To further clarify our analysis of what we call “substrate modes,” which are central to several of Reviewer 3’s comments, we have included four additional figures in this response (Response Figures 1-4). We have also included Response Figs. 1 & 3 in our revised manuscript as new Extended Figure 4. We believe these visualizations will help explain the superconducting mode’s spectral features and our corresponding fitting analysis, originally presented in Fig. 2 of the main manuscript.

Below are outlined questions and comments to the authors.

1. The authors mentioned the existence of parasitic modes in the system, but they provide little analysis and explanation of this parasitic mode. What is the exact origin of these modes, and how do they affect the device performance?

Response. We have combined our responses to Comment 1 and Comment 2. Our response and actions taken are presented below with Comment 2.

2. The aforementioned “parasitic modes” later become the “substrate mode” in the writing, yet there’s little explanation provided in the main text about it. What’s the difference between a parasitic mode and a substrate mode?

Response. The terms “parasitic” mode and “substrate mode” refer to the same modes in our system, and the same physical phenomenon: electromagnetic resonances supported by the sapphire substrate and surrounding packaging. The substrate and packaging act as a cavity, which supports additional resonances that are distinct from that of the superconducting cavity. These substrate/packaging-supported modes hybridize with the resonance of the superconducting cavity, thereby obfuscating the superconducting cavity’s spectral response in our main text Fig. 2.

Furthermore, this hybridization complicates the system dynamics; the substrate modes act as loss channels for photons in the superconducting cavity. This additional loss results in two primary affects:

1. Coupling between the superconducting cavity and the substrate modes reduces the intracavity photon number (intracavity power) of the superconducting cavity.
2. The hybridized mode has increased mode volume compared to the bare superconducting mode volume, thereby reducing the electro-optic coupling rate (g_0).

This loss can be understood analytically via the coupled-mode-theory equations of motion for our system, which are presented in detail in Supplementary Information Section 1.3. By including coupling between the superconducting cavity resonance and N substrate modes, we arrive at the following expressions for the linewidth, “ κ ,” and the cavity frequency detuning, “ Δ ,” of the bare superconducting resonance:

$$S_{21} = 1 - [\kappa_e/2]/[i(\Delta+\delta^-) + (\kappa+\gamma^-)/2]$$

Importantly, we observe two terms in the denominators of the above expressions, highlighted accordingly in yellow. These terms, which emerge as a direct consequence of the superconducting cavity resonance coupling to and hybridizing with the substrate modes, clearly lead to an increase in the bandwidth of the superconducting cavity resonance (i.e., $\kappa+\gamma^-$ increases), and a shift in the resonant frequency of the cavity (i.e., $\Delta+\delta^-$ decreases). Importantly, the stronger the coupling to the substrate modes, the greater the bandwidth/loss of the superconducting cavity resonance. This increase in loss from the superconducting cavity is what we refer to as the “parasitic” behavior of the substrate modes.

Action taken. To improve clarity, we exclusively use the phrasing “substrate mode(s)” in the revised manuscript. We have provided a detailed discussion of these effects, their origins, and a presentation of the above expressions, in the manuscript methods section, *Methods: Substrate mode modeling & analysis* and Extended Figures 3 & 4. Please see our revised manuscript Eqs. M6-M8, which coincide with the above expressions. Additionally, in *Methods: Inferring the electro-optic coupling rate g_0* we discuss and provide expressions on how the substrate modes impact our estimation of g_0 .

3. The authors should specify the exact section in the Method and Supplementary Information that readers should consult, rather than pointing to the entire document.

Response. We agree that adding these references will improve the manuscript's readability and presentation.

Action taken. We now specify all references pointing to the exact methods section (with hyperlinks) and supplementary sections.

4. The data and fitting in Fig. 2e are not convincing. With multiple peaks/dips in the data, the “multi-parameter model” is not reliable. “The substrate modes by transparent red vertical lines” hardly match the peaks.

Response. The multi-parameter model is built on the physical principle of a single, temperature-dependent superconducting mode hybridizing with multiple, temperature-independent substrate modes. The model's validity stems from its ability to accurately predict the expected physical behavior of the bare superconducting cavity (a smooth red-shift and broadening as temperature increases), which aligns with expectations from literature and theory. Our multi-parameter model enables us to extract this behavior from the full spectrum, in which the bare superconducting cavity mode hybridizes with the substrate-supported modes.

The red vertical lines in our original manuscript Fig. 2d depict the accurate locations of the *bare* substrate modes. The physical coupling/hybridization of the superconducting cavity to the substrate cavity leads to a shift between the bare mode resonant frequencies and the hybridized mode resonant frequencies. The red vertical lines depict the locations of the bare, *non-hybridized* modes. We have clarified this in the updated figure caption of our new submission. This behavior is well-captured by our model, as explained above in our response to comment 2.

To build intuition for our model, the figures below show how the observed mm-wave spectrum at two different operating temperatures is constructed by progressively adding the influence of each substrate mode to the predicted bare mm-wave cavity resonance (Response Fig. 1 & 2, below). The coupling strengths between the superconducting cavity resonance and the various substrate modes, as well as the resonant locations of the bare substrate modes and their linewidths as fit by our model, are given in Extended Table 1 of the manuscript. Note that in the last panel, “ $N_{\text{sub}}=8$ ” of Resp. Figures 1 & 2, the vertical dashed lines correspond to the *bare* frequency locations of the substrate modes (red dashed line for the bare frequency of the mm-wave mode), and they do not necessarily align with the peak locations of the hybridized modes that appear in the spectrum. This is due to the nature of hybridization, which we have discussed above in our response to comment 2, as well as in our response to Reviewer 3's comment 5.

The final calculated temperature sweep (Response Fig. 3), which includes the combined effects of all substrate modes, closely reproduces the features seen in the experimental data.

Finally, note that we are using the same (temperature-independent) substrate mode frequencies and linewidths to fit the data. The temperature change is only affecting the on-chip mm-wave mode. This increases our confidence in the model.

Action taken. We have revised Figure 2d,e in the manuscript to better illustrate our analysis. The redesigned Fig. 2d now shows the full raw data and a comparison of the raw data used for our multiparameter fitting with the extracted bare superconducting cavity response as a function of temperature. Additionally, Fig. 2e shows three line cuts of the colormap data presented in Fig 2d. This new presentation clearly demonstrates how our model predicts the behavior of the bare superconducting cavity resonance, separately from the full, substrate-mode-hybridized spectrum. The confusing red vertical lines denoting the bare substrate mode frequencies have also been removed for improved clarity. We believe the revised figure and corresponding text now provide a much clearer explanation and compelling justification for our modeling.

Additionally we have revised the section *Methods: Substrate mode modeling & analysis* further clarifying the impacts of the substrate modes with a new extended figure, Extended Figure 4.

Response Figure 1: A calculation showing the cumulative effect of adding substrate modes (N_{sub}) to the model at a base temperature of 4.89 K, using equations M4-M6 in the original manuscript (Eq. M6-M8 in the revised manuscript). The plot for $N_{\text{sub}}=0$ shows the simple Lorentzian lineshape of the bare (not hybridized) mm-wave cavity. As each substrate mode is included at particular frequencies (indicated by vertical dashed lines), the spectrum becomes progressively more complex. Importantly, note that the coupling between the superconducting cavity resonance and each substrate mode, leads to shifts in resonance frequency (as in Response Eq. 1). Therefore, the resulting Lorentzian dips in the spectrum do not align with the

bare, non-hybridized superconducting and substrate modes. This physical phenomenon is the reason why the red vertical lines (now removed) did not correspond to the dips in Fig. 2 of the submitted manuscript. This physical behavior of hybridized systems is well-established in the field and in literature.

Response Figure 2: The same calculation as in Response Fig. 1, but at a higher temperature of 6.20 K. The bare mm-wave cavity (ω_{RF}) has red-shifted due to increased quasiparticle population, leading to a different pattern of hybridization with the fixed-frequency substrate modes.

Response Figure 3: A colormap of the fully simulated transmission spectrum versus temperature, including the effects of all eight substrate modes. This simulation closely reproduces the complex avoided crossings and non-monotonic amplitude changes observed in

the experimental data (Fig. 2d in the revised manuscript), confirming the validity of our physical model.

5. In Fig. 2d, why do the modes' frequencies not drift continuously with temperature? Rather, they seem to “hop” from one vertical straight line to another. The amplitude of a certain line also changes non-monotonically. These need explanation.

Response. This is an insightful question that relates to the fundamental nature of hybridized modes. The apparent “hopping” and non-monotonic amplitude changes are characteristic signatures of so-called *avoided-crossings* between the temperature-dependent superconducting mode and the temperature-stable substrate modes. As the superconducting mode smoothly decreases in frequency with increasing temperature, it passes through fixed frequencies of the substrate modes. At each point of near-frequency degeneracy, the modes hybridize, causing the spectral features to shift abruptly. The “smoothness” with which we observe these avoided crossings is also in part related to the number of different temperatures at which we take RF spectra.

The figure below (Response Fig 4) provides a clear, isolated view of this phenomenon. We calculate the spectral response of the temperature-dependent superconducting mode with just a single fixed-frequency substrate mode. The *avoided crossing* behavior is evident, where the resonance peak appears to hop from one frequency to another as a function of temperature, with an apparent hopping of frequency. The frequency difference resulting in the hopping is proportional to the coupling rate and we list all such rates in Extended Table 1.

Action taken. We have added an explanation of this hybridization behavior in the main text (lines 126-148 in the revised manuscript) and section *Methods: Substrate mode modeling & analysis*.

Response Figure 4: A calculation isolating the interaction between the temperature-tuning bare RF mode (ω_{RF} , dashed white line) and a single, fixed-frequency substrate mode (ω_{sub}) using equations M4-M6 in the original manuscript (Eq. M6-M8 in the revised manuscript). The line cuts (top) and colormap (bottom) clearly show the characteristic avoided crossing, which explains the "hopping" and non-monotonic behavior seen in the full experimental data.

6. How does the author estimate the coupling rate accurately in the presence of the substrate modes? Is it possible to filter out the substrate mode response? And would this unwanted hybridization to the substrate mode limit the theoretical transduction efficiency?

Response. It is not possible to experimentally filter out the substrate modes, so their effect must be included in our analysis. Our model, described in *Methods: Substrate mode modeling & analysis*, accurately accounts for these substrate modes by treating their combined influence as an additional frequency shift ($\tilde{\delta}$) and loss rate ($\tilde{\gamma}$) for the superconducting cavity. By determining the parameters of the substrate modes via fitting our multi-parameter model (please see our responses to comments 4 and 5 above), we can calculate $\tilde{\delta}$ and $\tilde{\gamma}$ and determine the superconducting cavity's intracavity RF photon number, $n_{c,RF}$, in the presence of these substrate modes. With this accurate intracavity RF photon number, we can then properly infer the electro-optic coupling rate, g_0 (please see Eq. M8 in the original manuscript or Eq. M10 in the revised manuscript). Additionally, the reviewer is correct that this hybridization limits the theoretical transduction efficiency. It does so by increasing the total loss of the superconducting mode (reducing the intracavity photon population) and by altering the interacting RF mode's field distribution, which can reduce its overlap with the optical mode.

Action taken. We have updated the manuscript to more clearly describe these two mechanisms of how the substrate modes impact the device performance (please see lines 126-148, 184-192). We have also added an explanation about how one could mitigate/reduce the number of substrate modes in future devices (please see lines 126-139). Additionally, we have included details in the discussion section and *Methods: Substrate mode modeling & analysis, Methods: Inferring the electro-optic coupling rate g_0 .*

7. In the heterodyne measurements used in Fig. 2, the authors stated that these measurements could lead to the determination of total coupling rate and loss rate. However, no further information about this is provided. How to infer the coupling rate using the heterodyne measurement result?

Response. We thank the reviewer for requesting more detail on this technique. In the self-heterodyne measurement, a weak optical sideband is swept across the cavity resonance. The transmitted signal is detected on a fast photodiode, and a vector network analyzer (VNA) measures the amplitude and phase of the resulting beat note. This complex-valued VNA response is directly proportional to the cavity's transfer function, from which we can fit and extract the total loss rate (κ_j) and the external coupling rate ($\kappa_{e,j}$).

Action taken. We have added citations [38, 39] in the main text for further reading about this technique. We have also included a comprehensive description of the measurement setup and analysis procedure in the Supplementary Information sections S4.2 and S4.2.1.

8. In the transducer operation section, the term “laser piezo voltage” needs clarification. Does this refer to a piezoelectric actuator that controls the laser frequency, or does it serve another function in the measurement?

Response. Reviewer 3 is correct, “laser piezo voltage” refers to the piezoelectric actuator within our tunable laser source, which is used to fine-tune the laser's output wavelength.

Action taken. We have corrected the imprecise language.

9. The author also mentioned an asymmetry of the coupling rate to the red and blue sidebands in Fig. 2a. But in Fig.2a, the red and blue sidebands seem similar in their signal intensity without a clear asymmetry. The author should clarify what asymmetry they are referring to. In addition, the author should also indicate the supporting information section that the reader should read.

Response. We thank Reviewer 3 for highlighting this; the asymmetry we are referring to was originally presented in Fig. 3a, the difference in power transduced into the red-detuned sideband versus the power in the blue-detuned sideband. In the figure, the blue sideband peak is measured at -72.25 dBm, while the red is at -74.35 dBm, a power ratio of ~ 1.62 . This is caused by slight differences in the properties of the red- and blue-detuned optical modes in the optical racetrack cavity. Our input-output model, which accounts for these differences, predicts a ratio

of ~ 1.68 , in good agreement with the measurement (please see Supplementary Information 1.1).

Action taken. We added clarifying text as well as a new Fig. 3a to highlight the asymmetry we are referring to. In the updated figure, we have provided annotations that clearly label the power in each sideband to demonstrate the asymmetry unambiguously. We also now reference the specific supplementary information section in which we present the detailed analysis of this behavior.

10. The authors attribute the non-linear response in the transduction efficiency vs pump optical photon (Fig. 3b) to the local heating of the sub-THz resonator even at the low-cooperativity limit. This raises a fundamental concern: if heating effects are already significant at the current efficiency levels, how can this device achieve high transduction efficiency when strong optical pumping is required for optimal performance? Furthermore, this device is fabricated on sapphire, which intuitively should provide a better heat dissipation channel; however, the heating effect is surprising and warrants further explanation. The author should provide a more detailed analysis of this.

Response. Thank you to the reviewer for raising this critical point. The performance-limiting heating we observe is not due to lattice heating in the substrate, but is instead caused by optically induced quasiparticle generation within the superconducting electrodes. Stray optical pump photons can be absorbed into the superconductor, breaking Cooper pairs, and thus increase the non-equilibrium quasiparticle population. This non-equilibrium quasiparticle density degrades the superconducting cavity's performance in a manner analogous to a physical temperature increase. While sapphire is a good thermal conductor, the bottleneck in this case is the rate of quasiparticle recombination, not lack of heat dissipation. In other words, the "heating" of the superconducting cavity is not thermal, but rather an increase of quasiparticles in the superconductor.

As for the transduction efficiency limit, both lattice heating and quasiparticle generation are common limiting factors in all superconducting cavity electro-optic devices. Typical approaches to this challenge involve using a pulsed optical pump, which allows for high peak power to drive the transduction process, followed by a recovery period that allows for the system to relax back to equilibrium before the arrival of the next optical pulse.

Action taken. We have substantially revised the manuscript to improve clarity on discussions surrounding "local heating". The section previously titled "Locally heating the sub-THz resonator" is now titled "Optically induced quasiparticle generation". This section has been revised to provide a clear physical explanation of the mechanism. Furthermore, in the discussion section (lines 313-322 in the revised manuscript), we highlight the use of pulsed-pump operation as a path towards high-efficiency operation in future devices. We also indicate possible approaches to optical coupling onto the chip that could reduce scattered optical pump light on the chip surface.

11. What is the typical quality factor (Q) of the sub-THz cavity? In Fig. 4, the Q seems to be $\sim 107/2 = 53.5$ at low optical pumping. Is this considered a high-Q sub-THz resonator? How about the optical Q of the TFLN racetrack cavity? The triply resonant process should benefit from a high Q from both the sub-THz cavity and the optical cavity.

Response. Indeed a triply-resonant electro-optic transducer benefits from all cavities having high quality factors. In our case, the electro-optic efficiency is proportional to the square of the optical quality factor, Q_{opt}^2 and linearly proportional to the RF quality factor, Q_{RF} . Our optical quality factors are around $Q_{\text{opt}} \sim 1\text{e}6$ for both the pump and the sideband modes.

As Reviewer 3 calculates, the RF quality factor is around $Q_{\text{RF}} \sim 50$. This is a low quality factor, limited by dielectric loss from the SiO₂ cladding used in our fabrication process. In future devices, the fabrication process would be modified to remove this cladding.

Action taken. We have revised the discussion section of the manuscript to underscore Q-factor limitations and our strategies for improving it in future designs, such as removing the lossy oxide cladding. Please see lines 277-284 in the revised manuscript.

12. In Figure 4, the authors present the linewidth and frequency shift due to the optical heating. The author uses an effective temperature to calculate the local heating of the device. What is the definition of the effective temperature? In addition to a rise in temperature near the electrode, how about the added noise to the system?

Response. The “effective temperature” (T_{eff}) is a concept we use to quantify the impact of the optical pump on the superconductor. The optical pump photons induce a non-equilibrium quasiparticle density ($n_{\text{qp,neq}}$) within the superconductor, in addition to the thermally equilibrated quasiparticle density ($n_{\text{qp,eq}}$) due to the cryostat’s base temperature, T_b . Our “effective temperature” does not discriminate between non-equilibrium and equilibrium quasiparticle density and assigns a temperature, $T_{\text{eff}} > T_b$, to explain the total quasiparticle density. This concept of “effective temperature” is well-established in superconducting literature (see citation [40] in the revised manuscript). We determine this effective temperature by creating a calibration curve of the superconducting mode’s frequency (or loss rate) vs. physical cryostat temperature (which we take to be roughly equal to the device/chip temperature), and then mapping the optically-induced frequency shift (or loss rate) to a corresponding temperature on that curve.

To systematically characterize the added noise (number of noise photons referred to the input of the transducer) of transducers such as ours, pulsed operation along with SNSPD detection is required. However, these methods are outside the scope of this work. In the low-cooperativity limit, the added noise in an upconversion process is dominated by the cooperativity: $N_{\text{add}} \sim 1/C$. The highest cooperativity we measure is $\sim 1\text{e}-5$, which implies an added noise of $N_{\text{add}} \sim 1\text{e}5$ in a direct conversion process.

Action taken. As described in our response to Reviewer 3’s comment 10, we have substantially revised the manuscript to clarify our definition of “local heating.” The section previously entitled

“Locally heating the sub-THz resonator” is now entitled “Optically induced quasiparticle generation.” We provide two additional equations in the main text that define the quasiparticle density as in literature (Eq. 3) and a model that relates the effective temperature to the intracavity pump photon number (Eq. 4). Additionally, we have redesigned Figure 4 to provide a clear and graphical illustration of how T_{eff} is determined and modeled in our system.

Reviewer #4 (Remarks to the Author):

Response: Thank you to reviewer 4 for their time and contributions to feedback on this work.

Reviewer #5 (Remarks to the Author):

This manuscript presents the experimental demonstration of a triply-resonant cavity electro-optic transducer operating at ~100 GHz microwave range, implemented via integration of a superconducting NbTiN sub-THz resonator and a TFLN optical racetrack resonator. The authors report a single-photon electro-optic coupling rate of 0.7 kHz and a transduction efficiency 0.82×10^{-6} . The work addresses important engineering and physical challenges in realizing microwave-to-optical transduction at relative high frequency (~100 GHz), and offers thorough characterization. Electro-optic transduction at millimeter wave range is crucial for many applications. Achieving direct transduction at such a high microwave frequency typically encounters lots of key challenges. Therefore this reviewer do think this work represents a significant step and recommends the publication of Nature Communications. Below are some comments that may help improve the manuscript:

We are grateful to Reviewer 5 for their positive assessment and for the constructive suggestions to improve our manuscript. We agree with all the points raised and have revised the manuscript accordingly. Below, we detail our responses.

1. I think using millimeter wave is probably more suitable than sub-terahertz. This work demonstrates transduction at ~100GHz microwave frequencies, which exactly hits the range of millimeter wave and is a bit far from terahertz.

Response. Reviewer 5 raises a good point, ~100 GHz is quite close in the widely accepted mm-wave regime (30-300 GHz), while it only just enters into the sub-terahertz regime (90 GHz/100 GHz - 300 GHz). Within the literature, some authors refer to 100 GHz as “mm-wave,” while others call it “sub-THz”. Following Reviewer 5’s suggestion, we have elected to call it “mm-wave.”

Action taken. We have revised the manuscript to replace “sub-terahertz” with “millimeter-wave”, where appropriate.

2. The authors here used a single cavity with different modes separated by the FSR. It seems the drawback would be that both up and down conversion will happen at this time? The authors may want to comment on this a bit in the manuscript..

Response. Thank you to the reviewer for pointing this out. We *do* observe both up- and down-conversion.

Action taken. We have added more explanation and discussion of this effect and its impacts on quantum operation of our device into the discussion section of the manuscript (please see lines 301-312 in the revised manuscript).

3. Following up on comments 2, in Figure 3a, one could also see that both up and down sidebands show up. Then it might be a bit confusing for the readers that the authors only draw the up-conversion process in the Figure 1a.

Response. We agree with the reviewer that this might be confusing and mis-leading.

Action taken. We have amended Fig. 1a & 1b to include the down-conversion process (via dashed lines and another red-sideband arrow, respectively).